

# Full-length transcriptome and targeted metabolome analyses provide insights into defense mechanisms of *Malus sieversii* against *Agrilus mali*

Chuang Mei[1,2], Jie Yang[1], Peng Yan[2], Ning Li[2], Kai Ma[2], Aisajan Mamat[2], Liqun Han[2], Qinglong Dong[1], Ke Mao[1], Fengwang Ma[1] and Jixun Wang[2]

[1] State Key Laboratory of Crop Stress Biology for Arid Areas/Shaanxi Key Laboratory of Apple, College of Horticulture, Northwest A & F University, Yangling, Shaanxi, China
[2] Institute of Horticultural Crops, Xinjiang Academy of Agricultural Sciences/Scientific Observing and Experimental Station of Pomology (Xinjiang), Ministry of Agriculture, Urumqi, China

## ABSTRACT

*Malus sieversii* is the wild progenitor for many cultivars of domesticated apple and an important germplasm resource for breeding. However, this valuable species faces a significant threat in the areas north of the Tianshan Mountains in China, by the invasion of *Agrilus mali*, a destructive pest of apple trees belonging to the family Buprestidae. Our preliminary study has has shown that there may be resistance to this insect in *M. sieversii* plants in the field, but the corresponding molecular mechanisms remain unclear. In this study, we compared the response of insect-resistant and insect-susceptible plants of *M. sieversii* to insect feeding using full-length transcriptome and targeted metabolome. 112,103 non-chimeric full-length reads (FLNC) totaling 10.52 Gb of data were generating with Pacific Biosciences SingleMolecule, Real-Time (PacBio SMRT) sequencing. A total of 130.06 Gb data of long reads were acquired with an Illumina HiSeq. Function annotation indicated that the different expressed genes (DEGs) were mainly involved in signal transduction pathway of plant hormones and in the synthesis of compounds such as terpenes, quinones, flavonoids, and jasmonic acid. Through targeted metabolome analysis resistant strains showed higher levels of trans-cinnamic acid, caffeine and ferulic acid after pest infestation. This study helps to decipher the transcriptional changes and related signaling paths in *M. sieversii* after an insect feeding, which lays a foundation for further research on molecular mechanisms of insect resistance in apples.

# INTRODUCTION

*Malus sieversii*, distributed around the Tianshan Mountains, is the wild progenitor of its kind (*Duan et al., 2017*). Its flowers, fruits, nutrients, and growth habits are rich in diversity (*Forsline et al., 2010*). These species offers unique genetic resources that can be used in apple breeding programs (*Ma et al., 2017*). The apple buprestid beetle (*Agrilus mali*) is a coleopteran wood-borer that causes fatal damage to apple trees. Its larvae, which

Corresponding authors
Fengwang Ma,
fwm64@nwsuaf.edu.cn,
fwm64@sina.com
Jixun Wang, ee_wjx@163.com

live underneath the bark of trunks and branches, tunnel through the phloem and xylem in irregular pattern resulting in separation of the phloem from the xylem and death of the affected parts of the tree. In severe cases, the entire plant dies (*Yi et al., 2016*). The insect was listed as highly injurious by the Chinese government because it had killed over 600 ha of apple trees (*Mei et al., 2016*). Therefore, it will be difficult and/or impossible to recover many unique genes and alleles coded by *M. sierversii* if it disappeared.

When plants are attacked by insects and other herbivores, changes in signaling pathways trigger changes in proteins and metabolites in an attempt to circumvent these stresses (*Barah & Bones, 2015*; *Xie et al., 2018*). Morphological, biochemical, and molecular defenses can be triggered through immunological processes. Endogenous signals generated by the damaged cells also play a key role in a plant's ability to perceive herbivores, for example, through calcium channels, phosphorylation, and metabolic pathways of jasmonic acid, shikimic acid, amino acids, and terpenes (*Howe & Jander, 2008*; *Liu et al., 2016*; *War et al., 2012*). SA and ethylene can also be induced by some insect species. When insects invade, JA increase gradually.

Attacks by insects induce a systemic response in plants, which is manifested by changes in the transcriptional expression of specific functional genes and the accumulation of certain secondary metabolites (*Zhuang et al., 2018*). However, there are few reports on the defense to signaling system in apples affected by buprestid beetles. TIFY proteins, which are critical regulators in the JA signaling pathway, are involved in defense and stress responses during plant development, and they play a major role in plant reactions to biotic and abiotic stress (*Ebel et al., 2018*; *Thireault et al., 2015*). The TIFY protein gene family and its subfamily members improve disease and insect resistance by regulating signaling pathways of plant hormones (*Thireault et al., 2015*; *Xia et al., 2017*).

With the development of high-throughput sequencing technology, especially SMRT sequencing of genomes and transcripts can be acquired more efficiently (*Chao et al., 2018*). Transcriptome profiling has become an effective means for studying plant responses to stress, and it has been used widely to interpret how plants react to biotic and abiotic stress in various species (*Deng et al., 2018*; *Ren et al., 2018*; *Zhu, Li & Zheng, 2018*). In our early research we found that shoots of resistant plants contain much higher polyphenols and tannins, and much lesser insect numbers than susceptible plant shoots (*Mei et al., 2018*). In addition our feeding studies showed that *Agrilus mali* feeding preference on the leaves of susceptible plants compared to resistant one (Fig. S1). So we decided to analyze the specific mechanisms involved in the process through full-length transcriptome technology. In this study, analyses of the transcriptome of shoot phloem from resistant and susceptible strains of *M. sieversii* were carried out using RNA-seq technologies aiming to identify differential genes that may be involved in plant-insect interaction.

## MATERIAL AND METHODS

### Plant materials

On June 25, 2017, 50 shoots from 10 selected trees (five resistant and five susceptible) were collected from the wild *M. sieversii* population located in Gailiangchang (N43°22, E83°34,

the altitude 1,280–1,450 m), Xinyuan County, Xinjiang Autonomous Region. Field experiments was approved by the research council of the Institute of Horticultural Crops, Xinjiang Academy of Agricultural Sciences (approval number: XAASHC2016.02.08). The selected trees (resistant and susceptible) have been proved to be stable in their insect-interaction traits for years. All the trees were aged 45–60 years old.

### Infection treatment

The apple buprestid beetles usually parasitize the phloem of branches. According to their life cycle, adults lay eggs in the phloem underneath the bark. The larvae begin feeding the follow spring. The damage caused by larval feeding peaks in July and lasts until September. Ten mature larvae (one larvae for per shoot) were inserted into each tree through a simulated pore canal that created by T shape cutting of phloem. After a period of ten days, phloem were sampled from the infested and uninfested parts of the resistant and susceptible trees after the areas from the parts >3 cm away from the original mechanical damage had healed. (The sampled spots with obvious infection were chosen). Samples were frozen immediately in liquid nitrogen and stored at −80 °C before RNA extraction. Samples were named as follows: Insect-feeding resistant plant (IFR): Insect-feeding susceptible plants (IFS); None-feeding resistant plants (NFR); None-feeding susceptible plants (NFS).

### RNA extraction, cDNA library construction, and Illumina sequencing

Total RNA was extracted from each sample using a RNA extraction kit (TissueLyzer; Qiagen, Valencia, CA, USA), according to manufacturer's instructions. RNA purity was checked using a Nano Photometer spectrophotometer (IMPLEN, CA, USA), degradation and contamination were monitored on a 1% agarosegel, and integrity was assessed by an Agilent Bioanalyzer Nano 6000. Three μg of RNA was taken from each qualified sample and mixed. mRNA was purified from total RNA using poly-T oligo-attached magnetic beads. The sequencing library was prepared with a NEBNext®Ultra$^{TM}$ RNA Library Preparation Kit (NEB, USA). First strand cDNA was synthesized using a random hexamer primer and M-MuLV Reverse Transcriptase (RNase H-). DNA polymerase I and RNase H were added sequentially into the obtained first strand cDNA to produce the second cDNA strand. The fragments in the library were purified using an AMPureXP system (Beckman Coulter, Beverly, USA). The obtained double-stranded cDNA was subjected to end repair, addition of an A-tail, and adaptor ligation, then AMPure XP beads were used for size selection and, finally, PCR was adopted for enrichment to construct a cDNA library. Library quality was assessed with an Agilent Bioanalyzer 2100 system. After qualification and quantification of the library, short-read sequencing of each treatment × phenotype combination was performed using the Illumina HiSeqX-ten platform for RNA-Seq analysis.

### Construction of single molecule real-time library and sequencing

PacBio RS II and Illumina sequencing were done on the same sample from the same site. We mixed equal amounts of RNA and sequenced them. A SMARTer$^{TM}$ PCR cDNA Synthesis Kit was used to get full-length cDNA of mRNA, which was then screened with BluePippin$^{TM}$ Size. Three cDNA libraries (1–2 k, 2–3 k, 3–6 k) were constructed. PCR was used again to amplify the selected full-length cDNA. Then, they were repaired at

the end, ligated with the SMRT dumbbell-type adapters, and cleaved by exonuclease. A secondary screening was performed using Blue Pippin to obtain the sequencing libraries. After the libraries were constructed, they would quantify accurately using Qubit2.0, and the fragment size of libraries was detected using Agilent 2100. Only those complying with a standard could be sequenced. Long-read sequencing was performed using the PacBio RS II platform.

## Preprocessing of SMRT reads

Reads of Insert (ROI) sequences were extracted using the following criteria: $>= 0$ full-pass reading of the polymerase and sequence accuracy $>= 75$. Readings below 50 bp in length were discarded. ROI sequences were divided into two categories: full length sequences (that included a $5'$ primer, a $3'$ primer, and a polyA tail) and non-full-length sequences. We used the ICE algorithm of SMRT Analysis (v2.3.0) for iterative clustering, which collected similar sequences (i.e., multiple copies of the same transcript) into the same cluster. The parameters used are the default parameters. We used quiver to correct consensus sequences of each cluster, which yielded high quality transcripts (HQ) with an accuracy >99% (*Korlach et al., 2010*). In this project, the low-quality consensus sequences obtained from each sample were corrected using the corresponding Illumina RNA seq data through proof read to improve sequence accuracy (*Hackl et al., 2014*). PacBio RS II and Illumina sequencing data are deposited at the National Center for Biotechnology Information (NCBI) database (NCBI Sequence Read Archive SRP199016).

## Function annotation and classification

The annotation of transcriptome sequences were performed using the following seven databases: Nr (NCBI non-redundant protein sequences), Pfam (Protein family), KOG/COG (EuKaryotic Ortholog Groups and Clusters of Orthologous Groups of proteins), Swiss-Prot (a manually annotated and reviewed protein sequence database), KEGG (Kyoto Encyclopedia of Genes and Genomes) and GO (Gene Ontology). The read numbers were transformed to FPKM (fragments per kilobase of transcript sequence per millions base pairs sequenced) value for gene expression quantification.

## Real-time quantitative PCR verification

We applied qRT-PCR technology to validate the results of the differential expression analysis. Insect-resistant related genes with stable and drastic expression differences were selected for expression verification. Total RNA from each sample was reverse-transcribed using a 5× All-In-One MasterMix (Cat# G492, Abm, Canada) kit to synthesize the first strand of cDNA. The expression data in the Illumina HiSeq sequencing results were extracted for comparative analysis. Specific primers were designed using Primer-BLAST online software, and the melting curve was drawn to confirm PCR specificity. The quantitative PCR reaction conditions were as follows: 94 °C for 60 s, followed by 45 cycles of 94 °C for 5 s, 60 °C for 15 s, and 72°C for 10 s. SYBR was adapted to premix EvaGreen Express 2× qPCR MasterMix (Cat# MasterMix-ES, Abm, Canada). The test was performed with apple MdTUB1 as the reference gene using LightCycler 96 (Roche, USA).

Three biological replicates were set for each group, and we used three technical replicates for each biological replicate. Please see Table S1 for the sequence of qRT-PCR primers.

## Detection of secondary metabolites

The plant material was stored at −80 °C and ground with a grinder(MM 400, Retsch) at 30 Hz for 1 min. 100 mg of the powder was extracted in 1.0 mL of methanol, during which the sample was rotated three times to mix the solvent. After the extraction, 12,000g was centrifuged for 15 min to obtain a supernatant. The obtained supernatant was blown dry with nitrogen gas in a 35°C heating mode, then, resuspended in a mixture of 30% methanol with vortexing, centrifuged at 12,000 g for 15 min, and the supernatant was collected and stored for analysis on an LC/MS. The content of secondary metabolites was detected by liquid chromatography-mass spectrometry with reference to the method of *Francescato et al. (2013)*, *Pan, Ruth & Wang (2010)* and *Wojakowska et al. (2013)*. In this experiment, all chemical reagents were of analytically pure or chromatographically pure preparations. Chemical standards are from BioBioPha (http://www.biobiopha.com/) and Sigma Aldrich (http://www.sigmaaldrich.com/united-states.html).

The analysis conditions were as follows: The sample extracts were analyzed using an LC-ESI-MS/MS system (HPLC, Shim-pack UFLC Shimadzu CBM30A, http://www.shimadzu.com.cn/; MS, Applied Biosystems 4500 Q TRAP, http://www.appliedbiosystems.com/). The analytical conditions were as follows: HPLC: column, Waters ACQUITY UPLC HSS T3 C18 (1.8 $\mu$m, 2.1 mm*100 mm); solvent system, water (0.04% acetic acid): acetonitrile (0.04% acetic acid); gradient program, 95:5 V/V at 0 min, 5:95 V/V at 11.0 min, 5:95 V/V at 12.0 min, 95:5 V/V at 12.1 min, 95:5 V/V at 15.0 min; flow rate, 0.40 ml/min; temperature, 40 °C; injection volume: 5 $\mu$l. The effluent was alternatively connected to an ESI-triple quadrupole-linear ion trap (Q TRAP)-MS.

For quantification of metabolites, standard solutions of different concentrations were prepared. Then mass spectrometry was used to analyze 0.1 ug mL$^{-1}$, 0.2 ug mL$^{-1}$, 0.5 ug mL$^{-1}$, 1.0 ug mL$^{-1}$, 2.0 ug mL$^{-1}$, 5.0 ug mL$^{-1}$, 10.0 ug mL$^{-1}$ standard solutions and samples.

## Correlation analysis of transcriptome and metabolome data

By combining the metabolomics and transcriptomics data through the KEGG metabolic pathway, differentially expressed/accumulated genes and metabolites involved in the same biological process (KEGG Pathway) can be identified quickly.

## Statistical analysis

All the experimental data obtained were analyzed using SPSS 17 software (SPSS, Inc. Chicago, 186 IL, USA) and indicated by means $\pm$ ($n = 3$) standard deviation (SD). Data were analyzed using One-way ANOVA and Duncan test at a significance level of $p < 0.05$.

## RESULTS

### Morphological and physiological differences between resistant and susceptible plants

Susceptibility to the *Agrilus mali* invasion, soluble sugar, polyphenol as well as tannin contents of *M. sieversii* is shown (Fig. 1) to illustrate the difference between resistant and susceptible plants. Greater contents for polyphenol and tannin were observed in resistant plant shoots compared to susceptible plants (Figs. 1A–1B). However infection rate (number of insects) and soluble sugar contents in the shoots of susceptible plants were significantly higher than resistant plants (Fig. 1D). As the important secondary metabolites in plants, polyphenols and tannin are closely related to the disease resistance of plants, also play an important role in plant-insect interaction. Therefore higher polyphenol and tannin contents in resistant plant shoots may be related to resistance of these plants to *Agrilus mali* invasion. In the later section, we will focus on pathways involved biosynthesis of secondary metabolites such phenolic acids and plant-pathogen interaction.

### PacBio RS II and Illumina HiSeq sequencing

Transcriptome sequencing was performed using Illumina HiSeq X ten and PacBioRS II. First, all RNA samples were tested with an Illumina HiSeq X ten, which resulted in a total of 130.06 Gb Clean Data and Q30 up to 93.38% (Table S2). Then, an equal amount of the 12 RNA samples were mixed together for PacBio library preparation and sequencing. The full-length cDNA of the poly(A) + RNA sample was standardized and SMRT-sequenced using the PacBio RS II platform. The number of sequencing cells was two for a 1–2 K library, and one each for the 2–3 K and 3–6 K libraries. A total of 122,103 full-length non-chimeric (FLNC) reads (Table 1) and 275, 586 ROI (reads of insert) (Table 2) were obtained. SMRT Analysis software was employed to cluster the full-length sequences, and a total of 64,692 consensus transcript sequences were generated. Non-full-length sequence correction produced 47,181 high-quality, full-length transcript sequences and 17,511 low-quality sequences, which were corrected using Illumina Hiseq data (Table S3).

The sequences were clustered iteratively using SMRT Analysis (v2.3.0) software and the ICE (Iterative Clustering for Error Correction) algorithm. In this experiment, we received a total of 64,692 F01 consensus sequences with HQ (High-quality) and LQ (low-quality) transcripts that were obtained from different libraries (Table S3). At least 27,230 HQ transcripts were obtained from the 1–2 kb library, and 9,895 HQ were obtained from the 2–3 Kb library. In addition, the LQ consensus sequences were corrected by Illumina RNA seq platform using proof reads software to improve their accuracy (*Hackl et al., 2014*).

### Verification of RNA-Seq gene expression

To confirm the reliability of the RNA-Seq data, synthetic pathways, transcription factors, and important genes related to plant defense were screened from the annotation library of RNA-Seq DEGs data, and 18 of them were selected for real-time quantitative PCR (qRT-PCR). By doing so, effects of infection on expression of defense-related genes, and the accuracy of RNA-Seq data were verified (Fig. 2). Quantitative results showed that all 18 genes responded to infection intensively, which included 13 up-regulated and

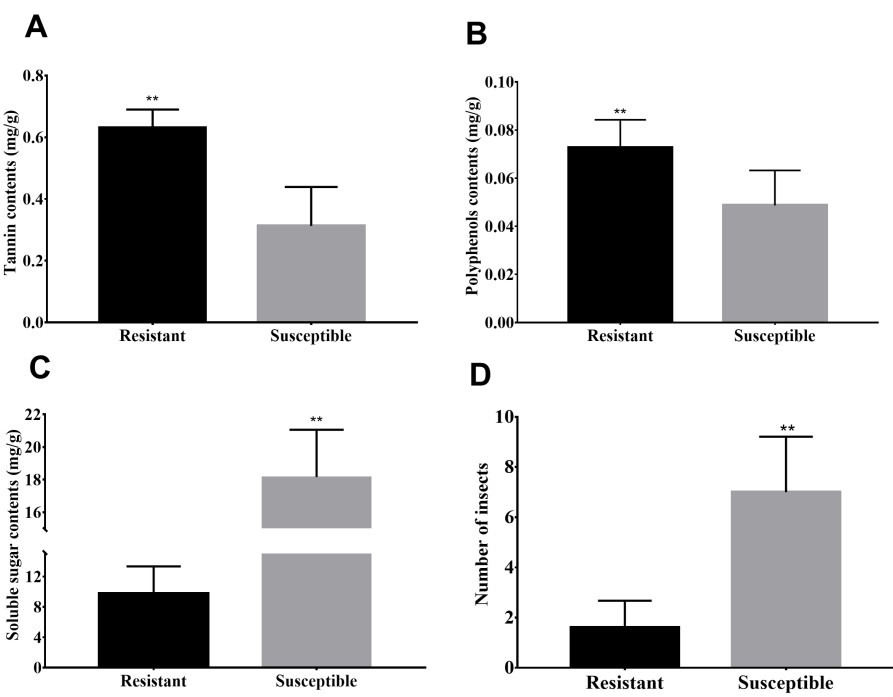

**Figure 1** **Differences in biochemical and infection rates between resistance and susceptible strains.**
(A–C) Spectrophotometric detection of the contents of soluble sugars, tannins, and polyphenols in the phloem of branches. (D) Number of bugs parasitic on resistance and susceptible strains. In this experiment, 10 branches with a length of 50 cm in each strain, then peel off the phloem and count the number of bugs under the bark.

**Table 1** **PacBio RS II sequencing data for samples collected from healthy and infested wild apple trees in China.**

| cDNA size | SMRT Cells | Polymerase Reads | Number of full-length reads | Number of full-length non-chimeric reads | Number of full-length non-chimeric reads | Full-Length Percentage (FL%) |
|---|---|---|---|---|---|---|
| 1–2 K | 2 | 300,584 | 72,228 | 71,914 | 71,914 | 48% |
| 2–3 K | 1 | 150,292 | 26,110 | 26,038 | 26,038 | 37% |
| 3–6 K | 1 | 150,292 | 24,165 | 24,151 | 24,151 | 45% |

Notes.
cDNA Size, the length of fragment in cDNA library; SMRT Cells, the amount of cells in all libraries; Polymerase Reads, the number of reads which were assembled by polymerase; Full-Length Percentage (FL%), the percent of full-length reads in ROI.

five down-regulated genes. Most changes in gene expression were consistent with the expression patterns in RNA-Seq results, which confirmed the reliability of Illumina HiSeq sequencing in this experiment. Furthermore, expression level of a gene annotated as protein TIFY9 increased significantly after insect induction, which exhibited more than a 30-fold up-regulation in resistant strains and a more than 10-fold up-regulation in the control. This indicated the active participation of *TIFY* genes in the regulation of an insect-resistant response.
**Table 2  ROI sequences for samples collected from healthy and infested wild apple trees in China.**

| cDNA size | Reads of insert | Read bases of insert | Mean read length of insert | Mean read quality of insert | Mean number of passes |
|---|---|---|---|---|---|
| 1–2 K | 151,994 | 292,511,929 | 1,924 | 0.93 | 13 |
| 2–3 K | 70,072 | 263,160,232 | 3,755 | 0.9 | 7 |
| 3–6 K | 53,520 | 250,555,242 | 4,681 | 0.88 | 5 |

**Notes.**
cDNA Size, the length of fragment in cDNA library, including primer, poly A and cDNA; Reads of Insert (ROI), the amount of ROI; Read Bases of Insert, the amount of bases in ROI; Mean Read Length of Insert, the quality average of ROI; Mean Number of Passes, the number of average sequencing times of ROI.

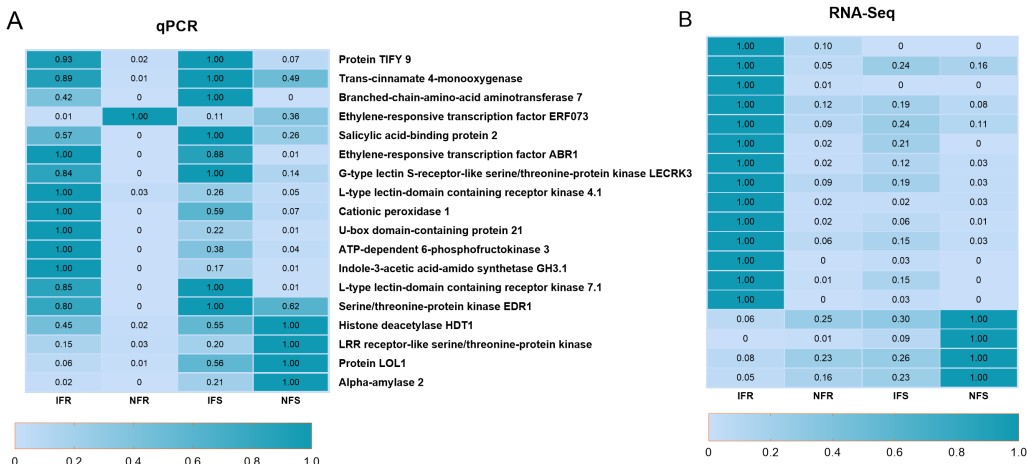

**Figure 2  qRT-PCR used to verify expression of biotic stress-related genes in Xinjiang Wild Apple.** (A) qRT-PCR data. (B) FPKM values obtained from the RNA-Seq data. Heat maps were produced using standardized figures that were transformed to a value between 0.0 and 1.0 by Min-Max normalization method.

## Functional annotation and selection of candidate DEGs related to plant pest resistance

In order to identify genes respond to *Agrilus mali* invasion in resistant plants, firstly we select common DEGs in NFS vs NFR (A), IFS vs IFR (B), NFR vs IFR(C) and NFS vs IFS (D) comparison groups, respectively. Then we used these common DEGs to build the Venn-diagram (Fig. S2). It is evident from the Venn graph that 266 DEGs respond in resistant genotype compared to the susceptible one and the control (non-feeding) samples(Fig. 3A). These DEGs mainly enriched in the pathways of phenylalanine metabolism, degradation of ammonic acids, Flavonoid biosynthesis, phenylpropanoid biosynthesis, terpenoid backbone biosynthesis, degradation of aromatic compounds, fatty acid metabolism, phosphatidylinositol signaling, plant hormone signal transduction pathway, plant-pathogen interaction etc. (Fig. 3B). Among these DEGs 240 of 266 were up-regulated and 26 DEGs were down-regulated in resistant plants after infection by *Agrilus mali* indicating that these genes may participated in plant-insect interaction (Fig. S3). In addition majority of these genes enrichen in secondary metabolic pathways verifying the important role of secondary metabolites in plant defense system. Among these 266

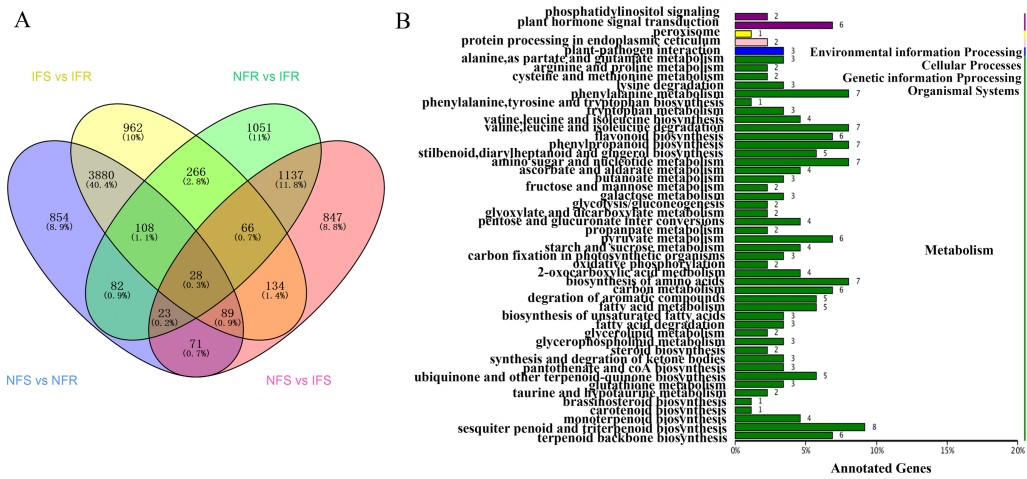

**Figure 3** **Screening and annotated functions of pest stress-responsive DEGs of Xinjiang Wild Apple.** (A) NFS vs NFR normal refers to the comparison of healthy parts between the control and insect-resistant strains (blue). IFS vs IFR pet refers to the comparison of infected site between the control and insect-resistant strains (yellow). NFR vs IFR presents the comparison between healthy. parts and infected sites of insect-resistant strains (green). NFS vs IFS refers to the comparison between healthy parts and infected sites in the control (pink). (B) KEGG classification of the 266 differentially expressed Unigene. The ordinate is the name of the KEGG metabolic pathway, and the abscissa is the number of unigene annotated to the pathway.

DEGs, we mined 19 DEGs that enriched in phenylalanine metabolic pathway, flavonoid biosynthetic pathway and plant-pathogen interaction. These selected 19 genes will be further analyzed in following section.

## Correlation analysis of transcriptome and metabolome

Ten days after insect feeding, we analyzed a group of bioactive secondary metabolites such as phenolic acids, alkaloids, flavones, salicylic acid, chlorogenic acid and benzoic acid that involved in plant defense system (Fig. S4). Among these metabolites, only 3 phenolic acids differentially accumulated in different samples (Figs. 4A–4C). Trans-cinnamic acid and caffeic acid content increased with 1.3-fold, 1.1-fold and 7.3-fold, 2.8-fold in susceptible and resistant plants, respectively. The fold changes of these two metabolites were much higher in resistant plant than in susceptible plant, indicating that these metabolites may play critical role in plant-insect interaction. Through correlation analysis, we found that genes encoding enzymes (PAL, C4H) responsible for biosynthesis of these four phenolic acids were differentially express in susceptible and resistant plants (Fig. 4D). In accordance with metabolites content, expressions of correlated genes were much higher in resistant plants compared to susceptible plants.

## DISCUSSION

The plant defense response to insect damage involves relatively complex regulatory pathways. Under biotic stress, stress response of plants can be understood through transcriptome analysis and identification of relevant candidate genes (*Hettenhausen et*

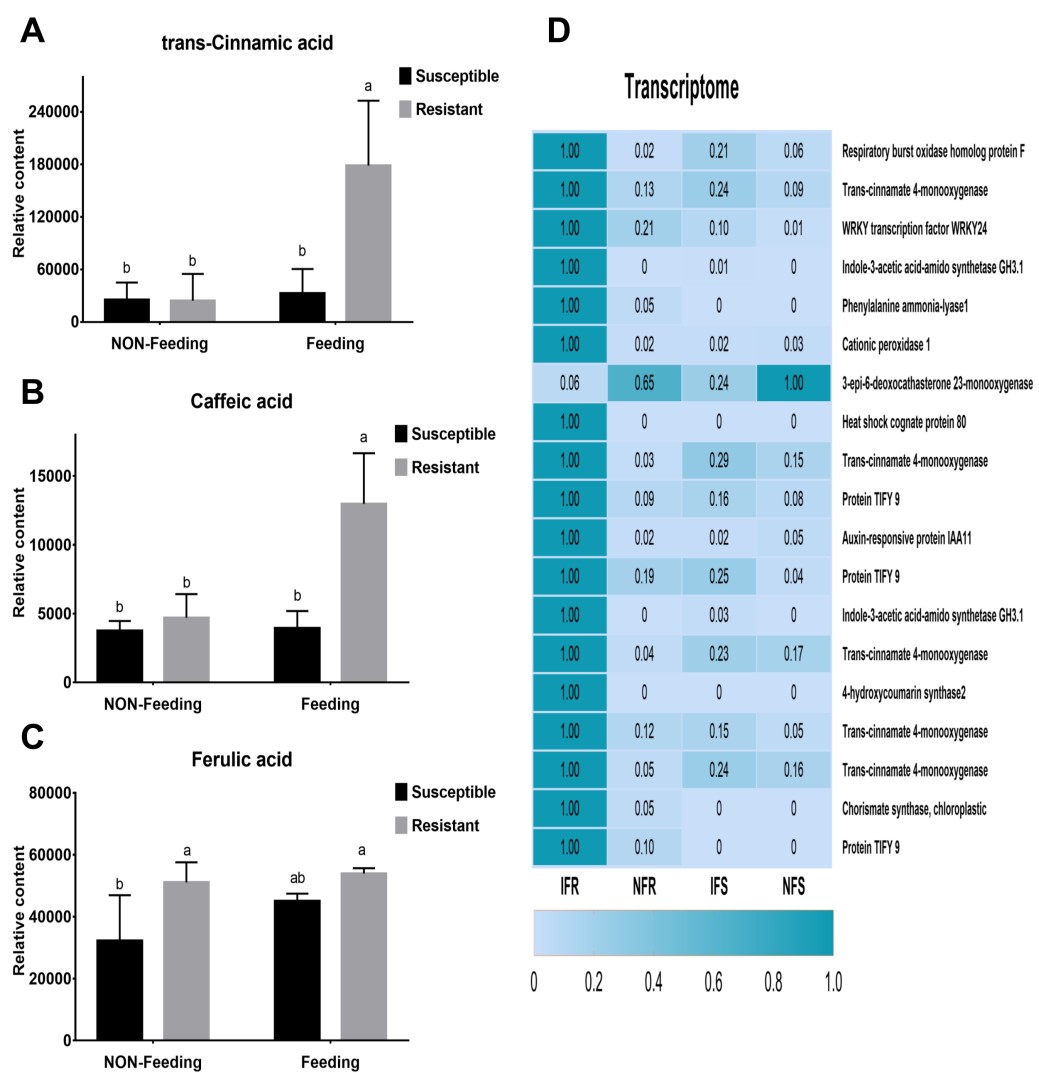

**Figure 4 Corresponding anti-insect metabolic pathways and DEGs after *Malus sieversii* infection with *Agrilus mali*.** (A–C) Insect-resistant metabolites detected in targeted metabolome, such as trans- cinnamic acid, caffeic acid and ferulic acid. Non-Feeding represents uninfested samples, Feeding represents infested samples. (D) Heat map of 19 insect resistance gene expression changes, and the 19 genes were selected from 266 DEGs. Heat maps were produced using standardized figures that were transformed to a value between 0.0 and 1.0 by Min-Max normalization method.

al., 2017; Zhuang et al., 2018). To obtain a higher-quality read length, we introduced real-time SMRT and corrected the results by combing Illumina HiSeq data to improve the accuracy of transcriptome analysis. Using four cells, 10.52 Gb clean data were obtained, which included 275,586 ROIs and 122,103 FLNC sequences (Tables 1 and 2). Illumina HiSeq sequencing data were used to correct 17,511 low-quality sequences (Table S3). Among the 9,598 DEGs, the genes (266) respond to *Agrilus mali* infection in resistant plants were mainly analyzed (Table S4). In present research we found that resistant plants contain significantly higher polyphenols and tannins than susceptible plants (Fig. 1). In

addition in our metabolomics we identify three phenolic acids (trans-cinamic acid, caffeic acid and ferulic acid) that differently accumulated in resistant and susceptible plants. All of these three metabolites were significantly increased in both two samples after *Agrilus mali* infection. Interestingly, fold changes of trans-cinamic acid and caffeic acid in resistant plants were much higher than in susceptible ones after *Agrilus mali* infection indicating their importance in plant-insect interaction (Fig. 4). Among these 266 DEGs we mined 19 genes that involved in secondary metabolism. Further correlation analysis between DEGS and metabolites revealed that majority of these 19 genes encode enzymes responsible for synthesis of above mentioned phenolic acids verifying the importance of these phenolic acids in plant defense system.

Phenylalanine lyase (PAL) catalyze phenylalanine ammonia to form cinnamic acid. it is reported that insect attack induce higher expression of *PAL*, accompanied by changing related hormone levels and metabolites (*Chaman, Copaja & Argando, 2003*; *Han et al., 2009*). In accordance with these findings, in our experiment we found that *PAL* was strongly induced by *Agrilus mali* infection and correspondingly trans-cinamic acid content were also increased exponentially. After entering the phenylpropane metabolic pathway, *PAL* further mediates the synthesis of flavonoids, lignin, anthocyanins, plant antitoxins and plant hormones that regulate plant-insect interaction. Cinnamate 4-hydroxylase (C4H), belonging to CYP73 family of P450 super family, responsible for catalyzing trans-cinamic acid to form *p*-coumaric acid. Studies have shown that plant-mediated RNAi silencing of the C4H monooxygenase gene significantly increased the tolerance of the larvae to gossyphenol, and the larvaes grew slowly when feeding on C4H over expressed plant materials (*Mao et al., 2007*). Similarly we identified five *C4H* genes that strongly induced in resistant plants compared to susceptible plants. C4H activity requires molecular oxygen and NADPH supplied by the NADPH generation system to produce activity. Plant NADPH oxidase, also known as *RBOHs*, is responsible for the production of signal ROS in the plant defense response (*Marino et al., 2012*). In *Arabidopsis* over expression of *RBOHD* and *RBOHF* increased RBOH/NADPH oxidase dependent ROS production and signaling in *Arabidopsis* immunity (*Torres, Dangl & Jones, 2001*). In our study we found that one transcripts of *RBOHF* up-regulated. In our study, *RBOHF* gene expression was up-regulated with 50-fold and 3.3-fold respectively in both resistant and susceptible plants after infection suggesting that *RBOHF* might mediate ROS signaling in response to external stimulus and activate plant defense system.

Jasmonate is an important plant hormone, which is necessary for acclimation of plants to adversity condition. JAZs transcription factors, belongs to a larger family of plant specific TIFY proteins, respond to the jasmonoyl-L-isoleucine (JA-Ile), which each members have different in response patterns to injury induction (*Chung & Howe, 2009*; *Thireault et al., 2015*). In this experiment, we identified three transcripts of *TIFY9* and all of these three genes were up-regulated after insect infection in both resistant and susceptible plants. TIFY family genes are important regulatory genes for JA accumulation, so it is speculated that TIFY proteins play an important role in plant-insect interaction. However functions of TIFY family members are still unclear and further study need to be done in the future works.

## CONCLUSIONS

This study shows that the contents of polyphenols, tannins, and soluble sugars in *M. sieversii* resistant strains are significantly different from those in the susceptible strains. Furthermore, 266 DEGs associated with insect-resistance were screened from transcriptome (240 expressions were up-regulated and 26 expressions were down-regulated). There are 19 accumulated DEGs associated with insect resistance in the phenylalanine metabolic pathway, flavonoids and jasmonate biosynthetic pathway. Among them, C4H, RBOHF, PAL and other genes are involved in the regulation of phenylpropane pathway. Targeted metabolomics results also confirm this. After the pest feeding, the trends in the secondary metabolites such as trans-cinnamic acid, caffeic acid and ferulic acid were consistent with the expression of the above genes. Therefore, it is likely that genes such as C4H, RBOHF, and PAL mediate phenylpropane signaling and induce the synthesis of downstream secondary metabolites, resulting in the insect-resistant phenotype.

## ACKNOWLEDGEMENTS

We would like to thank Thomas A. Gavin, Professor Emeritus, Cornell University, for help with editing this paper.

### Funding

This research was funded by the National Natural Science Foundation of China (31960583, 31701894) and the China Agriculture Research System (CARS-27). The funders had no role in study design, data collection and analysis, decision to publish, or preparation of the manuscript.

### Grant Disclosures

The following grant information was disclosed by the authors:
National Natural Science Foundation of China: 31960583, 31701894.
China Agriculture Research System: CARS-27.

### Competing Interests

The authors declare there are no competing interests.

### Author Contributions

- Chuang Mei conceived and designed the experiments, performed the experiments, analyzed the data, prepared figures and/or tables, authored or reviewed drafts of the paper, and approved the final draft.
- Jie Yang and Qinglong Dong analyzed the data, authored or reviewed drafts of the paper, and approved the final draft.
- Peng Yan performed the experiments, authored or reviewed drafts of the paper, and approved the final draft.

- Ning Li and Aisajan Mamat analyzed the data, prepared figures and/or tables, and approved the final draft.
- Kai Ma and Liqun Han performed the experiments, prepared figures and/or tables, and approved the final draft.
- Ke Mao, Fengwang Ma and Jixun Wang conceived and designed the experiments, authored or reviewed drafts of the paper, and approved the final draft.

## Field Study Permissions

The following information was supplied relating to field study approvals (i.e., approving body and any reference numbers):

Field experiments approved by the research council of the Institute of Horticultural Crops, Xinjiang Academy of Agricultural Sciences (approval number: XAASHC2016.02.08).

## Data Availability

The raw sequencing data of PacBio RS II and Illumina platforms are available in BioProject at NCBI: SRP199016. Raw data can also be found in Table S4, Data S1, Data S2 and Data S3.

## Supplemental Information

Supplemental information for this article can be found online at http://dx.doi.org/10.7717/peerj.8992#supplemental-information.

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
