# Peer review of "Full-length transcriptome and targeted metabolome analyses provide insights into defense mechanisms of Malus sieversii against Agrilus mali"

_PeerJ, doi:10.7717/peerj.8992_

## Round 0.1 · original submission · Major Revisions

Our reviewers identified several areas of concern, especially regarding the methods description and whether your comparisons were performed between healthy plants and infected plants or between diseased and healthy parts of the same plant. Please address all of their concerns and improve the discussion and the explanation of the rationale behind your choice of genes to focus on.

Reviewer 1 ·

Basic reporting

In this paper, authors reported the identification of genetic and metabolic factors involved on the resistance mechanisms of Malus sieversii (a wild relative of domesticated apple) to the insect Agrilus mali. I think, overall, the article provides some insight into this important issue, although several methodological aspects should be clarified in the manuscript. The structure of the paper is adequate but, in my opinion, the discussion section requires an extensive editing. The metabolomic analysis is not clearly justified and does not seem to provided any new information. Authors declare that they perform a target analysis but it is not clear to me on what type of molecules they are focusing. The figures are very difficult to evaluate due to the poor quality and the lack of figure legends.

Line 412: What is PBD?. In general, acronyms should be introduced the first time used.

Experimental design

I have several questions/comments and concerns about the methodology used.
1. The experimental design is not clearly explained. As far as I understood authors used two sets of plants (Resistant and Non-resistant) and performed two types of treatments (Infested and Control). Since authors named the non-resistant plant as control (as they did for the control treatments) the results are very confusing, and it is not clear to which comparison authors refer in each case. An example of this possible misinterpretation can be found in the line 402.
2. It is not clear to me what authors mean by biological replicates. It seems they called biological replicates to different samples collected on the same tree after infestation. If this is the case, these samples cannot be considered biological replicates.
3. Authors established that a Fold Change ≥ 2 was used as threshold to select DEGs. With this criterion, only up-regulated transcripts would be selected. The limit to detect down-regulated transcripts should be indicated also.
4. The LC-MS analysis is poorly explained. I cannot understand why, after the extraction, samples were dried and then redissolved in 30% methanol. What is advantage of the reducing the methanol content?. The parameters and equipment used for the mass spectroscopy analysis must be also described.

Validity of the findings

The objective of the paper is clear and well stated and the overall methodology used is adequate. However, how the results improved our knowledge about the resistance mechanism against this pest is barely explained. Most of the information provided in the discussion section must be reallocated to the results section. The discussion section must include only an overall view of the results integrating the results from the different analysis performed and comparing them with those found in the literature. The resistant role of the TIFY genes in apple is not fully discussed. There is not clear conclusion in the manuscript.

Reviewer 2 ·

Basic reporting

1. English writing needs to be much improved, such as wrong grammars, many reduplicate and redundant descriptions.
2. Others comments see blow as general comments for the author.

Experimental design

1. In experiment design, it seems that the samples of healthy and insect-infection parts were collect from the same plants. As far as I know, many herbivores induced systemic defense response, so I am wondering the gene expression in the “healthy parts” of the infested parts may also be challenged. Therefore the screened genes associated with infestation may be inaccurate.
2. In targeted metabolome analyses, the authors only analyses the metabolites in resistant plants, how about the susceptible plants? Based on our experience, herbivore infestation may also affect the secondary metabolism of susceptible plants. Thus the change of some metabolites may be a stress response of the plants, not the “resistance response” of the resistant plants. I suggest to carry out metabolome analyses of the susceptible plants for fully comparison analyses.

Validity of the findings

1. The data were poorly analyzed, and more details should be given regarding how to analyze the data. For example, the authors just list the data acquired by omics approaches without further explanation and discussion. In addition, in the materials and methods part, there were no statistical analysis methods regarding the accumulation of secondary metabolites, but the figure 7 showed some statistical significances. Please aid detail methods.
2. The data were not properly explained in discussion. For example, in L420-428, the authors just list some related results, but not deeply discuss the current results.

Additional comments

Specific comments:
1. L24 There should be a blank in “M.sieversii”, please check through the whole text.
2. L27 Please give the full terms of “SMRT”.
3. L179 The cutoff of the significance of GO and KEGG analyses?
4. L188-189 The results of TIFY family genes should be placed in results part.
5. L204-205, Table S8? TableS3?
6. L207 The samples for metabolome analyses are same to transcriptome??
Please give mere details.
7. L276 The cutoff of Fold Change “≥2 or ≤0.5 and FDR <0.01”?
8. L365 The unite of the chemicals was “ug/ml”?
9. L392-394 This information should be placed in results part.
10. L412 “PBD”?
11. L434-435 “22 genes in the TIFY domain”? I only found 16 genes in figure 5.
12. The Figures legends should be clear and concise, please give more details.
1) Figure 1, I suggest to name the samples with the same treatment with uniform titles like Figure 6A.
2) Figure 3A, “RvsR”? ““CvsC””?
3) Figure 4, graph symbol should be “Resistant”?
4) Figure 7, the statistical methods? Replications? Error bar is SE or SD?
13. The legend of TableS8 in the main text and the supplemental materials is not consistent.
14. For reference, there are many inconsistent styles. Please check reference style cite in the main text and reference list carefully. L175, “(Yangyang et al. 2006)”; L217, “Xiangqing et al. 2010”; L587, “Liu X, 2, Mei W, Soltis P, 3, Soltis D, 2,3, and Barbazuk W, 3. 2017.”
15. …

Reviewer 3 ·

Basic reporting

Many missing details in the methods section and there was not much substance to the discussion section. The authors mostly restated their findings and loosely connected their results to what has been observed in previous systems, but their ideas were not well-linked with these previous studies. I believe this section should be rewritten entirely to improve its impact.

In addition, the quality of the analysis of the differentially expressed genes is extremely poor. There is no detail about what software was used for the analysis and the authors do not clearly describe how they selected the 266 DEGs for further analysis (they point to a Venn Diagram, but don't explain why this part of the Venn Diagram is relevant to their study).

Experimental design

There are many details of the design that are not clear and I question the validity of the metabolite data since it does not appear that the authors sampled data from infested and uninfested susceptible and resistant trees (only two bars are shown in the graph and it is not clear which treatments are included because the figure legends are missing). There is no statistical analysis described for the qPCR results and it is also not clear if they used the same samples that were used for the RNA-Seq analysis.

Validity of the findings

Very hard to assess because there are some key details missing in the methods section that are really needed to properly evaluate the design and the results.

Additional comments

The manuscript by Mei et al presents a transcriptome assembly for Malus sieversii and a differential expression analysis of susceptible and resistant plants before and after infestation by Agrilus mali, a devastating pest of apples in China. Overall, the authors have generated a new genetic resource that can be used to understand responses to biotic and abiotic stresses in apples and that potentially can be used to mine genes associated with insect resistance that can be used to generate more resilient cultivars of apple.

Although the resource will be useful, the experimental methods are lacking numerous details that are needed to evaluate the quality and validity of the analysis. The figures are confusing and are missing legends and it is unclear how the authors selected differentially expressed genes that are associated with insect-resistance (they point to a Venn Diagram, but fail to show the logic behind why they selected these particular DEGs). The quality of the results presented are also poor, mostly because it is difficult to say why the authors chose to focus on this particular set of genes, but additionally, the analysis of the TIFYs is unclear (they mention that the TIFYs that are identified show 1:1 orthology with those annotated in the apple genome, but they don’t show any analysis to support that).

Finally, the discussion section 5-6 previous studies and only loosely connects the findings from this study to those from previous studies. About 75% of the discussion section is repeated from the results section.

Line 20: What do you mean that when you say this is the original species of domesticated apple? Is it the wild progenitor for many cultivars of domesticated apple? Please be more specifc.
Line 21: However, this valuable species faces a significant threat in the areas north of the Tianshan Mountains in China, by the invasion of Agrilus mali, a destructive pest of apple trees belonging to the family Buprestidae.
Line 23: has shown that there may be resistance to this insect in M. sieversii plants in the field
Line 24: change insect-sensitive to insect-susceptible
Line 26 (and throughout): change insect infection to insect feeding
Line 26: 112,103 non-chimeric full-length reads (FLNC) totaling 10.52 Gb of data were generating with PacBio SMRT sequencing.
Line 28: the phrase “short-read long sequencing data” doesn’t make sense in this context. Please rephrase.
Line 29: Were these alternative splicing events and lcRNAs observed in response to insect feeding or were they just documented in the in the transcriptome assembly?
Line 29: Functional annotation indicated
Line 31: and in the synthesis of compounds such as terpenes….
Line 32: Since TIFY genes are important
Line 33: in apple species in general or just in this apple species? Be more specific.
Line 34: Here and throughout, change Totally to “In total,”
Line 34: in the apple genome….Also, please specify which apple genome you are referring to.
Line 37: Through targeted metabolome analysis….showed higher levels of….p-coumaric acid
Line 47: Please explain what you mean when you say it shares high “homogeneity” with cultivated apples.
Line 48: Here and throughout the manuscript, there are numerous places where there are spaces missing between the references and the last word of a sentence. Please carefully review the manuscript and correct this.
Line 48: What do you mean when you say it is the most primitive of its kind? This phrase is ambiguous.
Line 49: Likewise, this statement also lacks context and is ambiguous. Its flowers, fruits, nutrients, and growth habits are rich in diversity….please elaborate what you mean here because it is not clear how the the flowers and fruits produced by this tree can be rich in diversity.
Line 49: These species offers unique genetic resources that can be used in apple breeding programs.
Line 51: Its larvae, which live underneath the bark of trunks and branches, tunnel through the phloem and xylem in a spiral pattern, resulting in separation of the phloem from the xylem and death of the affected parts of the tree.
Line 57: Therefore, it will be difficult and/or impossible to recover many unique genes and alleles coded by M. sierversii if it disappeared.
Line 57: This statement is not entirely true. Some apples can self-pollinate (ie, not all are self incompatible) and self-incompatibility does not necessarily mean that is the species is doomed if it is threatened by an insect pest. The species can still persist and reproduce through cross-pollination. Addition, the inclusion of the term “asexual reproduction” in this statement does not make much sense. How can this species be self-incompatible and only reproduce by asexual reproduction? Please check this and reword. Some apples can produce vegetatively, but this is not the same as saying that they can only produce by asexual reproduction.
Line 58: When plants are attacked by insects and other herbivores, changes in signaling pathways trigger changes in proteins and metabolites in an attempt to circumvent these stresses.
Line 61: Morphological, biochemical, and molecular defenses can be triggered through immunological processes.
Line 67: SA and ethylene can also be induced by some insect species.
Line 67: I don’t think this is the correct use of the word “attenuate.” Usually, JA levels initially increase in response to damage by herbivores. Attenutation is a reduction. Please revise thiss atement for additional clarity.
Line 69: Expression levels of several genes associated with JA and ethylene synthesis/response (???) pathways were increased/decreased?….and higher levels of salicylic acid….
This statement is a little confusing. What are you trying to say about the genes associated with JA and ethylene and how does that relate to what you are saying about the role of salicylic acid in insect defense and response?
Line 71: Change systemic resistant response to systemic response. The response does not always lead to resistance
Line 74: Change “signal” to signaling
Line 82: In a previous study of the TIFY gene family in apple….30 proteins were identified that contained the TIFY domain that were further classified into the following subfamilies:….
Line 87: and the accuracy of the gene predictions and annotations
Line 88: This improved reference genome will enhance our ability to study these gene families in other apple species.
Line 90: SMRT sequencing,
Line 94: How do you know that these “resistant” individuals represent different strains? Could they simply be individuals that are more resistant to the insect than other members of the population? Please provide some more details here.
Line 96: Pleas specific pest species here
Line 98: DEGs were identified between what? Infested and uninfested trees from the resistant cultivar? Infested resistant trees and uninfested susceptible trees? Please be clear what comparisons were made to identify the DEGs.
Line 99: change “screened out” to “identified.”
Line 109: Please revise this statement: We previously identified five individuals that grew at the same elevation (same elevation as what?) that seemed to have resistance against A. mali.
Line 109: Just because these trees did not appear to have insect damage does not indicate that they have resistance. Please describe in detail how you determined that they were indeed resistant to this insect.
Line 116: According to their life cycle, adults lay eggs in the phloem underneath the bark.
Lines 118-124: This section needs to be rewritten in reorganized for additional clarity. The larvae begin feeding the follow spring. The damage (due to what?...larval feeding or adults?) peaks in July. How can the damage last only until September? The damage is likely permanent? Do you mean that the larvae continue to feed until September? On July 25, 2017, five trees that had shown consistent resistance to A. mali over a 5-year period were infested with larvae as were five control trees (how were the control trees selected?). Mature larvae (how many) were inserted into each tree through a simulated pore canal (please describe how this canal was created). After a period of ten days, phloem was sampled from the infested and uninfested parts of the resistant and control trees after the area from the original mechanical damage had healed. Please describe here what regions were sampled. A picture as a figure or supplemental figure pointing to the regions that were sampled and diagraming how the trees were infested would be useful.
Line 127: Please rewrite this section into complete sentences.
Line 135: by an Agilent Bioanalyzer Nano 6000.
Line 141: an AMPure
Line 143: addition of an A-tail
Line 145: After qualification and quantification of the library,
Line 146: short-read sequencing of each treatment x phenotype combination was performed using the Illumina HiSeq X-ten platform for RNA-Seq analysis. Please specify what types of reads were obtained (150 x 150 PE reads or other)? Also, the X-ten platform is not really intended for transcriptome sequencing and was not supported by Illumina when this platformed was launched. Please explain why this platform was selected.
Line 149: Do you mean to say that you mixed the RNA from the control/resistant genotypes and the infected and uninfected sites together for long-read sequencing? Please rewrite these sentences for clarity.
Line 151: was used to construct full-length cDNA sequences. Please elaborate on how the BluePippin was used for size selection. Did you select one size only for the three libraries you sequenced? Or did you select different sizes for each of the three libraries? If so, please describe in detail how this was done.
Line 155: What was this secondary screening for? Quality check?
Line 156: What fragment size was selected for sequencing? Please elaborate by stating in detail what you mean when you say that only samples that complied with a standard were used.
Line 158: Long-read sequencing was performed using XXXXX.
Line 160: Please define ROI
Line 160: ROI sequences were extracted using the following criteria: >=0 full-pass reading of the polymerase and sequence accuracy >=75. Readings below 50 bp in length were discarded…
Line 165: Define the ICE algorithm and explain what parameters were used in the clustering protocol (ie, what values were changed from default parameters)?
Lines 173-179: Describe the blast searches and HMM searches that were performed to generate these functional annotations. E-value thresholds need to be provided as well as whether blastp/x was used. Versions and dates that databases were downloaded also need to be provided. Please also describe how GO, KOG, eggNOG and KEGG terms were obtained and please specify which transcripts were searched (all transcripts or just the ones that were predicted to code for proteins?).
Line 181: Why not search your transcriptome for putative TIFY domains using HMMER as opposed to searching known proteins that contain this domain against your transcriptome database? This may improve your ability to detect divergent TIFY domain proteins that may be dissimilar to those derived from the reference genome.
Line 192: to validate the results of the differential expression analysis.
Line 193: Please describe in more detail which genes you selected for validation and why.
Line 194: Total RNA from each sample was
Line 195-196: The last part of this sentence is not necessary. and we further detected the expression amount by real-time quantitative PCR.
Line 205: Please describe in detail how the qPCR data were analyzed. In addition, I don’t see any details in the methods about how the Illumina reads were mapped to the long-read assembly and how differentially expressed genes were identified. These procedures must be included.
Line 208: change “at an ultra-low temperature” to “at -80C”
Line 208: was extracted in 1.0 mL of methanol.
Line 209: during which the sample was rotated three times to mix the solvent.
Line 212: then resuspended in a mixture of 30% methanol with vortexing, centrifuged at 12,000 g for 15 minutes, and the supernatant was collected and stored for analysis on an LC/MS.
Line 213-214: This statement is not necessary. During redissolution, vortex isused to completely dissolve the target substance
Line 221: The analysis conditions were as follows: (please rewrite all of this into complete sentences). Please use past tense. It looks like somebody else wrote this section compared to the rest of the methods section.
Line 234-235: five resistant and five control trees
Line 239: and was sequenced on the PacBio RS II platform.
Line 240: and one each for the 2-3K and 3-6K libraries.
Line 244: Do you mean that of the 64,692 sequences, 47,181 were high-quality full length and the rest were low-quality after correction with the PacBio reads. These sequences were error corrected with the short-read Illumina data?
Line 254: What are F01 consensus sequences? Do you mean FL?
Line 257: According to your definition in the previous statement, the LQ reads came from the PacBio instrument and not the Illumina instrument. They were error corrected using the Illumina reads. Please correct.
Line 261: Long non-coding RNA (lncRNAs) is defined as RNA >200 nt that does not contain any open reading frames. To identify putative lcnRNAs, we assessed each transcript for coding potential.
Line 263-264: Please add details about these analyses to the methods section and clearly define the algorithms that were used to assess coding potential and define any parameters that were used in these programs.

Line 271: What are p1 and p2?
Line 273-275: No details regarding the programs that were used for differential expression analysis are supplied in the methods. Please add these details.
Line 278: Please explain which parts of the Venn Diagram were considered relevant for insect-resistance and describe your rationale for determining that these were relevant. For example, genes whose expression levels were upregulated or downregulated in the resistant plant relative to both uninfested resistant and infested and uinfested controls. Or something to that affect.
Line 286: How do these ~9500 DEGs differ from the 266-insect related DEGs that were described in the previous section. Are they just genes that are differentially expressed in all vs all pairwise comparisons? It would be more beneficial to separate them out by pairwise treatment comparisons because there are important details that are lost by presenting all of the DEGs together.
Line 292: Were these genes up/down regulated in insect-resistant samples? More details are needed here..
Line 310: expression level of a gene annotated as F01.PB7083
Line 318: change ‘developed’ to ‘were observed’
Line 319: You can’t be certain that these genes were located on certain chromosomes because there is no reference genome available for the species/cultivars that you analyzed in this study. It is possible that the gene order may have shuffled or that chromosomal translocations have occurred in this species compared to the reference genome. Please remove this section from the manuscript.
Line 321: Are these TIFY genes from your transcriptome assembly? This was also unclear in the methods section.
Line 323: These statements are unclear? What do you mean by TIFY groups? The subfamilies that were described earlier in the methods? If so, please be more specific and also provide more details about how the intron/exon structure differed between the different groups/families.
Line 330: Of course it is not surprising that you were able to find the TIFY domain in all of these proteins because the TIFY domain was used in your searches to identify these genes.
Line 339: There is not a sufficient level of detail in the methods section to evaluate the quality of this co-expression analysis. In addition, I do not believe there are a sufficient number of replicates or treatments for any meaningful results to be extracted rom this data.
Line 340: How were these eight TIFY genes chosen? Were they the ones that responded to insect feeding? In the section above, you mention a total of 16 TIFYs.
Line 345: Please explain how these TIFYs were named in your sample. Were they annotated based on their orthology with TIFY genes in the apple genome?
Line 346: Elevated expression levels were also observed in the control samples, although their levels were increased by only 13.9 and 6.9-fold in the infested control plants relative to the uninfested plants.
Line 354: More details are needed for this analysis.
Line 384: we performed long-read SMRT sequencing on the PacBio RS II instrument.
Line 389-390: The lncRNAs were simply annotated and their roles in response to insect infestation were not investigated in this study. Please remove this statement.
Line 414: In what plant species and in response to what insect? More details are needed here.
Line 425: There are some spaces missing in this sentence.
Line 427: that are involved in resistance.
Line 432-434: This statement does not connect well with the rest of the ideas presented in the paragraph.
Figure 4: Where are the statistical analyses for the qPCR data? In addition, there is no information about how the data were standardized in the methods section (was ddCT used for DE, how was relative expression level calculated)? What is normal vs Pet? Why not just call these infested and uinfested?
Figure legends appear to be missing for all figures and therefore, it is difficult to determine what is being presented in some of these figures. Sample labeling is also inconsistent across figures.
The raw PacBio and Illumina data need to be deposited into NCBI or some other suitable repository. The authors should also attempt to make their transcripts available for use

---

## Round 0.2 · Major Revisions

Please address the points noted by the reviewer, especially point 5. I agree with the reviewer that the 266 DEG are not what you claim to be: they seem to be the DEGs common to infected susceptible and resistant plants that, in resistant plants, are not DEG in the control state. The genes that respond to infection in both susceptible and resistant plants seem to be 23+28+66+1137. As an outsider I would probably expect the response to the resistance question to lie in the genes that are DEG in IFR vs IFS but not DEG in IFS vs. NFS comparison (i.e. 299 + 962), but I must confess that the interpretation of the Venn diagram is not very straightforward to me, and I think that in any case the reasoning behind your choice of quadrant should be made clearer and more explicit.

Reviewer 1 ·

Basic reporting

Although authors have made and effort to improve the manuscript, there are still few things that should be corrected. There are still many grammar mistakes in text. English writing needs to be improved.

Experimental design

The overall methodology used is adequate.

Validity of the findings

I think, overall, the article provides some insight into the plant-insect interaction field.

Additional comments

1- In M&M authors state that they used standard solutions to perform absolute quantification of metabolites, however in the figure 4 they reported relative content of phenolic compounds.
2- Authors should include information in M&M about how they perform the correlation analysis.
3- I can not see what information is reported in Figure 4D.
4- Line 276: what authors mean by “In our previous research”, they refer to the Figure 1 of a previous paper?.
5- Line 246: “It is evident from the Venn graph that 266 DEGs were respond to insect invasion in both resistant and susceptible plants”, according to the Venn diagram this sentence is not correct. The 266 DEGs respond in resistant genotype compared to the susceptible one and the control (non-feeding) samples.
6- A general discussion about the transcriptomic response must be added.

---

## Round 0.3 · Minor Revisions

I am afraid that your description of the choice of genes in the Venn diagram is still very confusing. Please consider reforming that portion of the write-up. To enable further analysis of your data by interested readers, please include the full data used to build the Venn-diagram (in the form of a spreadsheet with one row per gene and one column per comparison , with each cell stating whether the gene is DEG in that specific comparison or not)

Reviewer 1 ·

Basic reporting

In the actual form the paper is suitable for publication

Experimental design

In the actual form the paper is suitable for publication

Validity of the findings

In the actual form the paper is suitable for publication

---

## Round 0.4 · Minor Revisions

Thank you for providing the requested data. The high-quality final transcripts described in this work need also to be deposited at NCBI and the functional annotation of these transcripts needs to be provided before final acceptance.

---

## Round 0.5 · accepted · Accept

Thank you for depositing the annotations.